# Rapid reconstruction of neural circuits using tissue expansion and light sheet microscopy

**Joshua L Lillvis[1]\*, Hideo Otsuna[1], Xiaoyu Ding[1], Igor Pisarev[1], Takashi Kawase[1], Jennifer Colonell[1], Konrad Rokicki[1], Cristian Goina[1], Ruixuan Gao[1,2,3], Amy Hu[1], Kaiyu Wang[1], John Bogovic[1], Daniel E Milkie[1], Linus Meienberg[4], Brett D Mensh[1], Edward S Boyden[2,5], Stephan Saalfeld[1], Paul W Tillberg[1], Barry J Dickson[1,6]\***

[1]Janelia Research Campus, Howard Hughes Medical Institute, Ashburn, United States; [2]MIT McGovern Institute for Brain Research, Cambridge, United States; [3]Departments of Chemistry and Biological Sciences, University of Illinois Chicago, Chicago, United States; [4]ETH Zurich, Zurich, Switzerland; [5]Howard Hughes Medical Institute, Cambridge, United States; [6]Queensland Brain Institute, The University of Queensland, St Lucia, Australia

**\*For correspondence:**
lillvisj@janelia.hhmi.org (JLL);
b.dickson@uq.edu.au (BJD)

**Abstract** Brain function is mediated by the physiological coordination of a vast, intricately connected network of molecular and cellular components. The physiological properties of neural network components can be quantified with high throughput. The ability to assess many animals per study has been critical in relating physiological properties to behavior. By contrast, the synaptic structure of neural circuits is presently quantifiable only with low throughput. This low throughput hampers efforts to understand how variations in network structure relate to variations in behavior. For neuroanatomical reconstruction, there is a methodological gulf between electron microscopic (EM) methods, which yield dense connectomes at considerable expense and low throughput, and light microscopic (LM) methods, which provide molecular and cell-type specificity at high throughput but without synaptic resolution. To bridge this gulf, we developed a high-throughput analysis pipeline and imaging protocol using tissue expansion and light sheet microscopy (ExLLSM) to rapidly reconstruct selected circuits across many animals with single-synapse resolution and molecular contrast. Using *Drosophila* to validate this approach, we demonstrate that it yields synaptic counts similar to those obtained by EM, enables synaptic connectivity to be compared across sex and experience, and can be used to correlate structural connectivity, functional connectivity, and behavior. This approach fills a critical methodological gap in studying variability in the structure and function of neural circuits across individuals within and between species.

## Editor's evaluation

This article introduces a new light microscopy pipeline for imaging and fast reconstruction of the synaptic connections of individual neuronal types in the fruit fly and for correlated investigation of circuit structure, function, and behavior in the same animal. Because of its speed and accessibility, this approach enables the mapping of selected neuronal circuits of multiple animals across different conditions and behavioral states, thus filling an important gap in brain research.

## Introduction

Major efforts are underway to reconstruct comprehensive wiring diagrams of the nervous systems of various species. These connectome projects are fueled by recent advances in electron microscopy (EM) and automated image analysis methods and motivated by the idea that knowing the exact pattern of synaptic connectivity within a neural network is necessary, though not sufficient, to understand how it functions. Currently, most connectome projects are focused on generating reference connectivity maps for selected organisms (*White et al., 1986*; *Ryan et al., 2016*; *Eichler et al., 2017*; *Hildebrand et al., 2017*; *Zheng et al., 2018*; *Cook et al., 2019*; *Scheffer et al., 2020*; *Bae et al., 2021*). This current state of connectomics research is somewhat analogous to the state of genomics research just over two decades ago. Advances in DNA sequencing had made it possible to generate reference genomes for selected organisms, and ultimately also for humans. The success of these genome projects created the need to rapidly resequence targeted genomic regions across large numbers of samples in order to determine how these sequences vary from individual to individual and from species to species, so as to better understand how the genome 'works' and how it evolves. Similarly, with the expanding collection of reference connectomes, there is now an increasingly urgent need for methods that allow rapid but sparse reconstruction of neural circuits across large numbers of samples.

Electron microscopy, the method of choice for the reconstruction of dense reference connectomes, is not well suited to this task. The time and cost involved in EM reconstruction currently limit such comparative analyses to very small volumes (*Bourne and Harris, 2012*; *Bumbarger et al., 2013*; *Valdes-Aleman et al., 2021*) and to a few well-equipped laboratories (*Figure 1*). Moreover, because EM produces a dense image without molecular specificity, it is exceedingly difficult to highlight individual neurons and molecules. EM thus provides both too much and too little information for a typical comparative analysis of neuronal connectivity: too much, in that the specific neurons of interest first need to be identified amongst the vast tangle of processes revealed in the EM images, and too little, in that essential information on, for example, the chemical nature of individual connections is generally not available. Light microscopy (LM) provides the means to readily highlight the neurons and molecules of interest across many individual animals but has traditionally lacked the resolution needed to resolve individual synaptic connections (*Figure 1*). Recently, however, a combination of expansion microscopy (*Tillberg et al., 2016*) and lattice light sheet imaging (*Chen et al., 2014a*) (ExLLSM) has been shown to afford sufficient resolution to reveal single synapses and molecular labels and to have sufficient speed to image large volumes of neural tissue across many animals (*Gao et al., 2019*).

We aimed to develop a rapid, high-throughput and cost-effective ExLLSM pipeline to resample the connectivity of selected neural circuits with synaptic resolution and molecular and genetic specificity. Our primary goal was to establish a method to address variation in neural circuits. Variations in structural connectivity arise through evolutionary, ontogenetic, environmental, stochastic, and experience-dependent processes. A major challenge in neuroscience is to understand how such structural variations relate to species, sex, individual, and experiential differences in circuit physiology, function, and behavior. There is not necessarily a one-to-one mapping between circuit connectivity, physiology, and output (*Marder, 2011*; *Marder et al., 2015*). Thus, a full investigation of the structure–function relationships in neural circuits requires that each be examined across many individuals, ideally with structural and functional data acquired from the same samples. ExLLSM should be well suited to this task.

Here, we report the development of an ExLLSM circuit reconstruction method and pipeline, which we use to rapidly resample selected circuits within the *Drosophila melanogaster* connectome. We show that this method yields synaptic counts consistent with those obtained by EM, and that it can reveal structures such as electrical connections that are largely invisible to EM. Moreover, we apply our ExLLSM pipeline to

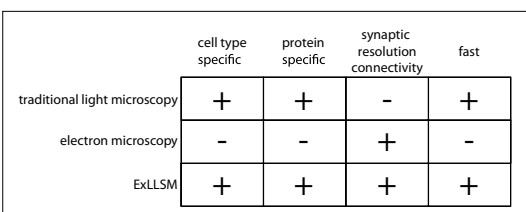

| | cell type specific | protein specific | synaptic resolution connectivity | fast |
|---|---|---|---|---|
| traditional light microscopy | + | + | - | + |
| electron microscopy | - | - | + | - |
| ExLLSM | + | + | + | + |

**Figure 1.** Features of traditional light microscopy, electron microscopy, and expansion microscopy and lattice light sheet imaging (ExLLSM). ExLLSM combines the synaptic resolution connectivity of electron microscopy with the cell-type specificity, protein specificity, and speed of light microscopy to allow the structural connectivity between targeted neurons to be quantified across many individual animals.

reveal state-dependent differences in neuronal connectivity and correlate structural, physiological, and behavioral data across multiple individuals.

## Results

### Data acquisition and analysis

*Drosophila* brains were dissected, in some cases from animals that had undergone prior behavioral and physiological analysis. Brains were isometrically expanded to 8× their original size with high mechanical stability via an interpenetrating network gel. No shrinkage or further expansion of these samples was observed across multiple-day imaging sessions. With the LLSM settings used, the central brain of *Drosophila* can be imaged in three colors in ~5 days at a resolution (~30 × 30 × 100 nm) sufficient to identify individual electrical and chemical presynaptic and postsynaptic sites across the brain.

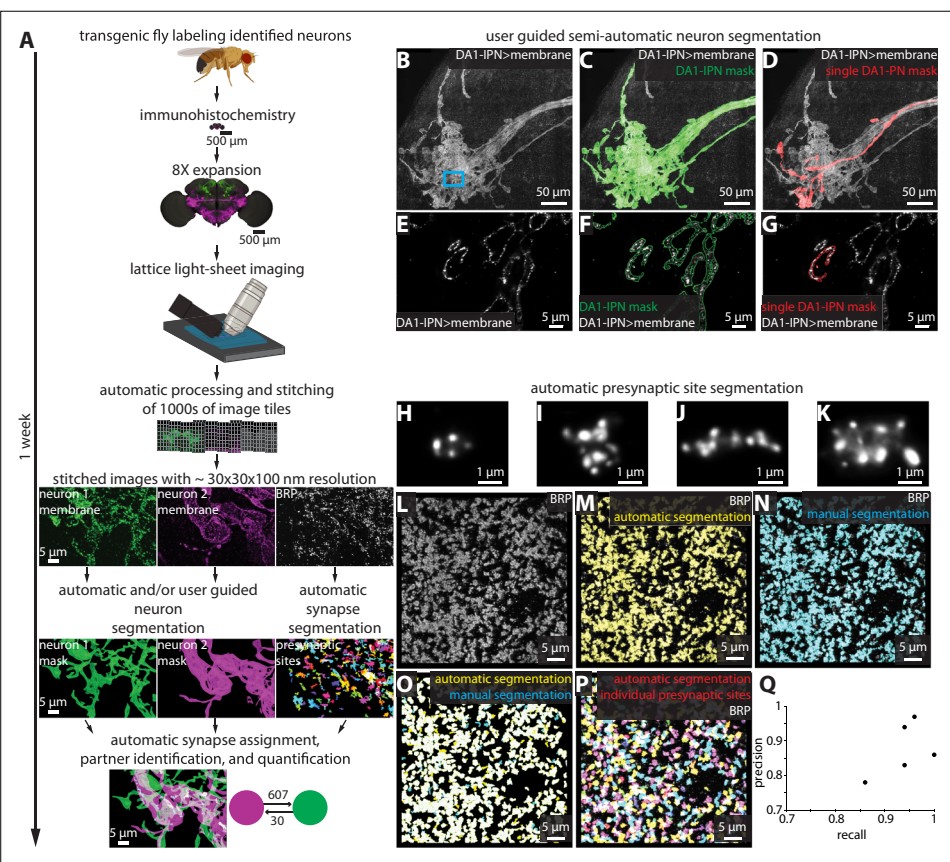

**Figure 2.** Largely automated ExLLSM image processing and analysis pipeline. (**A**) Overview. (**B–G**) Semiautomatic neuron segmentation. DA1-IPN neuron cluster (**B**, white) in the lateral horn with neuron masks generated by semiautomatic segmentation of the entire cluster (green, **C**) and manual segmentation of a single neuron (red, **D**). (**E–G**) Single z-slice of boxed region in (**B**) showing a cross section of presynaptic boutons. (**H–Q**) Automatic presynaptic site segmentation. (**H–K**) Examples of individual presynaptic site morphologies at 8× as visualized by labeling BRP. Crop from the optic lobe showing the BRP label (white, **L**) with automatically (yellow, **M**) and manually (ground truth, cyan, **N**) segmented presynaptic sites. (**O**) Overlay of automatic presynaptic site segmentation results (yellow) and ground truth data (cyan). (**P**) Overlay of BRP (white) and automatically segmented individual presynaptic sites (multicolor). (**Q**) Precision-recall plot of automatic presynaptic site segmentation. Results from four different brain regions of five independent samples.

The online version of this article includes the following figure supplement(s) for figure 2:

**Figure supplement 1.** VVD Viewer semiautomatic neuron segmentation and post-VVD mask processing.

**Figure supplement 2.** Automatic neuron segmentation.

**Figure supplement 3.** Workflows for automatic synaptic site detection, assignment, and partner identification.

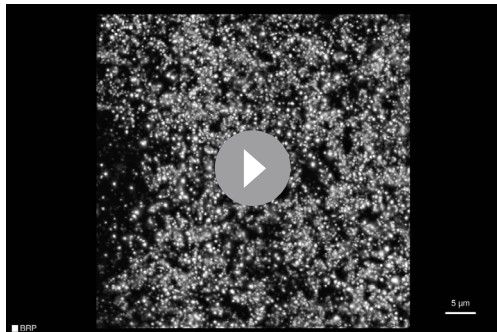

**Video 1.** Example of presynaptic site morphologies and distribution in the optic lobe medulla at 8× as visualized by labeling BRP and automatically segmenting individual presynaptic sites (multicolor).

https://elifesciences.org/articles/81248/figures#video1

To process and analyze the images obtained by 8× ExLLSM (hereafter referred to as ExLLSM), we developed a largely automated pipeline to quantify synaptic connectivity between genetically identified neurons (*Figure 2A*). The pipeline combines a previously published automated image tile processing and stitching method (*Gao et al., 2019*), with newly developed software to visualize multi-terabyte LM image volumes in 3D, semiautomatically (*Figure 2B–G, Figure 2—figure supplement 1*) and automatically segment neurons (*Figure 2—figure supplement 2*), detect and classify pre- and postsynaptic sites (*Figure 2L–Q, Video 1, Figure 2—figure supplement 3*), assign synaptic sites to segmented neurons, quantify the number, size, and location of neuronal connections (*Figure 2—figure supplement 3*), and to export these data as images and tables.

In developing this pipeline, we updated the VVD Viewer LM visualization and analysis software (*Wan et al., 2012*) to allow smooth 3D visualization and segmentation of large datasets. We also developed several Fiji plugins (*Schindelin et al., 2012*) and Apache Spark-based tools (*Zaharia et al., 2016*) to simplify and accelerate the processing and analysis of big image data. These plugins include tools for cropping, maximum intensity projection (MIP) creation, pixel intensity thresholding, signal cross-talk subtraction, 3D component connecting, component analysis, and component size thresholding.

We assembled these open-source tools into easy-to-use computational workflows via Nextflow (*Di Tommaso et al., 2017; Figure 2—figure supplements 1–3*). All of the tools are described in the 'Materials and methods' and on GitHub (*Lillvis et al., 2021*), where they are maintained along with user manuals and usage examples. The pipeline was built to analyze multi-terabyte ExLLSM images of the *Drosophila* nervous system. However, the only aspects of the pipeline that are specific to this data are the trained convolutional neural network (*Çiçek et al., 2016*) models used for synapse detection and neuron segmentation (*Figure 2L–Q, Video 1, Figure 2—figure supplement 2*). These models can be retrained on new data from other organisms or microscopes and seamlessly integrated into the pipeline.

## Presynaptic site counts obtained by ExLLSM match those from EM

To validate our data acquisition and analysis pipeline, we compared presynaptic site counts obtained via ExLLSM to those obtained via EM. In *Drosophila*, presynaptic active zones (hereafter referred to as presynaptic sites) are anatomically identified in EM images by T-bars (*Huang et al., 2018; Buhmann et al., 2021*), where synaptic vesicles are pooled and released. One component of T-bars is the Bruchpilot (Brp) protein (*Fouquet et al., 2009*), which can be detected in LM images through either ubiquitous (*Wagh et al., 2006*) or genetically restricted labeling (*Chen et al., 2014b*). We therefore used BRP as the marker to identify presynaptic sites of three distinct neuron types – optic lobe L2 neurons, antennal lobe DA1-IPN neurons, and ascending SAG neurons – and compared presynaptic site counts obtained by ExLLSM to T-bar counts obtained via EM.

Optic lobe L2 neurons make synapses onto motion detecting neurons in the medulla (*Takemura et al., 2013; Tuthill et al., 2013*). In an EM volume comprising seven optic lobe columns from one animal obtained by focused-ion beam milling scanning electron microscopy (FIB-SEM), an average of 207 presynaptic sites were detected per L2 neuron (*Takemura et al., 2015*). We used both the ubiquitous and genetically restricted strategies to label presynaptic sites in L2. For ubiquitous labeling (*Figure 2—figure supplement 3A*), we used the nc82 antibody to label BRP (*Wagh et al., 2006*) and a split-GAL4 driver line to specifically label the L2 neurons (*Tuthill et al., 2013; Figure 3A–G, Video 2*). We imaged large sections of the optic lobe medulla and segmented 10 individual L2 neurons in each of three flies, counting an average of 210 presynaptic sites per L2 neuron (*Figure 3C*). For genetically restricted labeling, we used the synaptic tagging with recombination (STaR) method (*Chen*

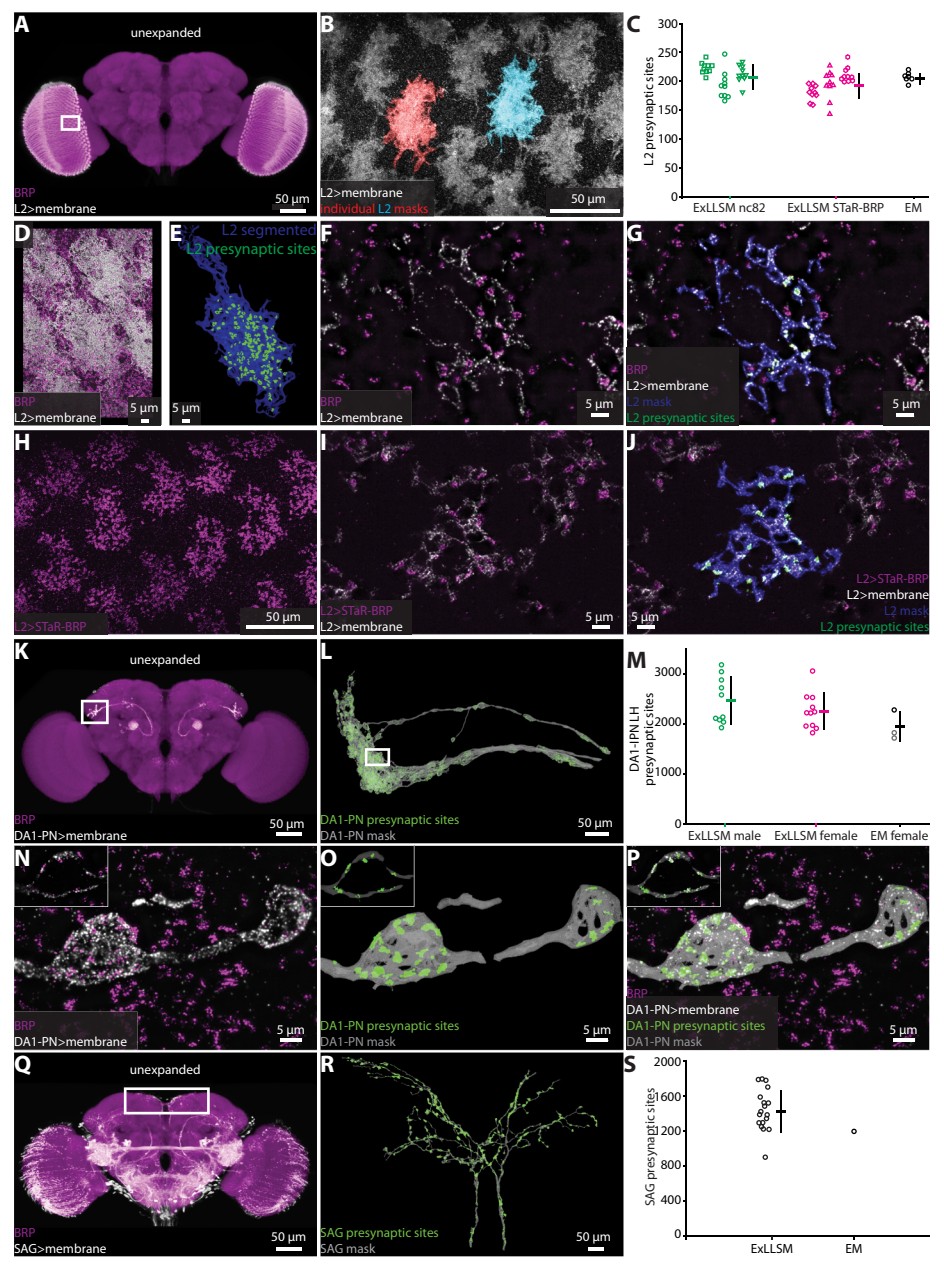

**Figure 3.** Automatic presynaptic site classification and neuron assignment. (**A, B, D–G**) Representative examples of quantifying presynaptic sites in L2 neurons via ubiquitous BRP labeling. (**A**) Unexpanded female brain with BRP (magenta) and the membrane of L2 neurons (white) labeled. (**B**) Representative 8× view of L2 neurons from rectangle region in (**A**), with two individual L2 neurons segmented. (**C**) Quantification of L2 presynaptic sites detected by ExLLSM using either ubiquitous (nc82) or restricted (STaR-BRP) labeling (n = 10 neurons, in each of the three samples for both) and by the presence of T-bars in EM (n = 7 neurons in adjacent columns). (**D**) Group of L2 neurons and BRP. (**E**) Mask of single segmented L2 neuron from (**D**) with automatically identified L2 presynaptic sites. (**F**) Zoom in on five z-slices from (**D**) with analyses overlaid (**G**). (**H–J**) Representative example of quantifying presynaptic sites in L2 via STaR-BRP labeling. (**G**) STaR-BRP expression in a group of L2 neurons. BRP is only labeled in L2 neurons. (**I**) Zoom in on five z-slices from (**H**) with individual L2 mask and automatically identified L2 presynaptic sites overlaid (**J**). (**K, L, N–P**) Representative example of quantifying presynaptic sites in DA1-IPN neurons via ubiquitous BRP labeling by the nc82 antibody. (**K**) Unexpanded female brain with BRP and the membrane of DA1-IPN neurons labeled. (**L**) 8× view of segmented DA1-IPN neurons in the lateral horn, approximately from rectangle region in (K) with automatically detected DA1-IPN presynaptic sites labeled. (**M**) Quantification of DA1-IPN lateral horn presynaptic sites. Neuron cluster grouped across 11 female and 10 male

*Figure 3 continued on next page*

*Figure 3 continued*

ExLLSM samples, and 2 electron microscopy (EM) samples (FIB-SEM in black, left and right TEM in gray). (**N**) Zoom in on raw signal of two DA1-IPN presynaptic boutons from rectangle region in (**L**). Inset shows a single z-slice. (**O**) Analyzed data from (**N**). (**P**) Overlay of raw data from (**N**) and analyzed data from (**O**). (**Q, R**) Representative example of quantifying presynaptic sites in SAG neurons via ubiquitous BRP labeling by the nc82 antibody. (**Q**) Unexpanded female brain with BRP and the membrane of SAG (and off-target) neurons labeled. (**R**) 8× view of segmented SAG neurons from rectangle region in (**Q**) with automatically detected SAG presynaptic sites labeled. (**S**) Quantification of SAG presynaptic sites in 18 ExLLSM samples and 1 FIB-SEM sample. (**C, M, S**) Individual samples and mean ± SD are plotted.

*et al., 2014b*) to specifically label the endogenous BRP protein in L2 neurons (*Figure 3H–J*). With this method, we counted an average of 195 synapses, also across a total of 30 L2 neurons (*Figure 3C*). Thus, the L2 synapse counts obtained by 8× ExLLSM were similar with both labeling methods and matched the EM counts (*Figure 3C*).

The DA1-IPN neurons comprise a group of 7–8 sexually dimorphic neurons that relay conspecific odors from the DA1 glomerulus in the antennal lobe to the mushroom body and lateral horn (*Marin et al., 2002*; *Wong et al., 2002*; *Stockinger et al., 2005*; *Jefferis et al., 2007*; *Kurtovic et al., 2007*; *Kohl et al., 2013*; *Figure 3K–P*). We labeled these neurons using a LexAGAD driver line, and imaged, segmented, and masked their projections in the lateral horn. We counted presynaptic sites in 11 females and 10 males (*Figure 3K–P*), and compared these counts to those obtained from one hemisphere of the FIB-SEM female hemibrain (*Scheffer et al., 2020*) and both hemispheres in the transmission electron microscopy (TEM) volume of another female (*Zheng et al., 2018*; *Bates et al., 2020*). On average, the female ExLLSM presynaptic site count of 2251 was similar to both EM datasets (FIB-SEM hemibrain: 2274; TEM: 1837 left, 1735 right) (*Figure 3M*). Although the average count in males (2466) was higher than in females, this difference was not statistically significant (*t*-test, two-tailed p=0.25).

The bilaterally paired female-specific SAG ascending neurons relay the fly's mating status (virgin or mated) to the central brain (*Feng et al., 2014*). In 18 females, we labeled SAG via a LexAGAD driver line, segmented the SAG neurons from the off-target neurons, and counted an average of 1440 presynaptic sites (*Figure 3Q–S*, *Videos 3 and 4*). In the FIB-SEM female hemibrain (*Scheffer et al., 2020*), the SAG neurons have 1198 presynaptic sites, which falls within the range of counts we obtained by ExLLSM (*Figure 3S*).

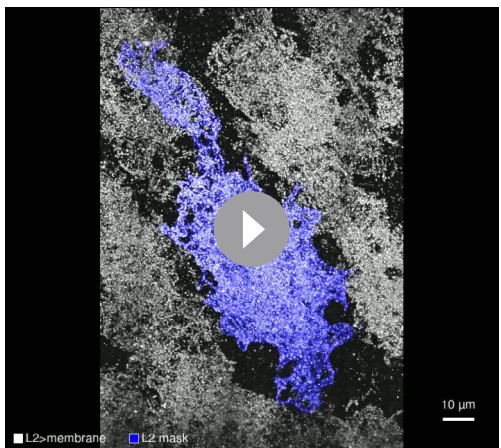

**Video 2.** Representative example of automatic presynaptic site identification in a single L2 neuron. L2 neurons were labeled by a membrane reporter. A single L2 was semiautomatically segmented and an L2 mask was generated. Presynaptic sites labeled by BRP were automatically classified and assigned to L2.
https://elifesciences.org/articles/81248/figures#video2

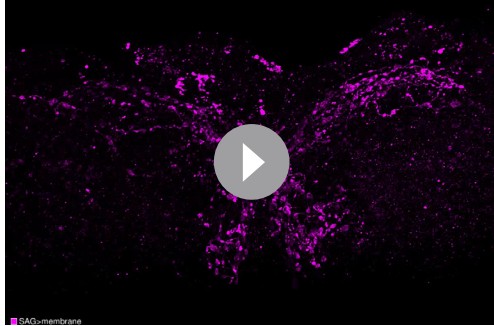

**Video 3.** Representative example of identifying SAG to pC1 connections via pC1 postsynaptic site contact with SAG presynaptic sites. SAG was labeled by a membrane reporter. SAG was semiautomatically segmented, and an SAG mask was generated. Presynaptic sites labeled by BRP were automatically classified and assigned to SAG. pC1 postsynaptic sites labeled by Drep2-HA specifically in pC1 were automatically classified. Connections between pC1 postsynaptic sites and SAG presynaptic sites were automatically identified.
https://elifesciences.org/articles/81248/figures#video3

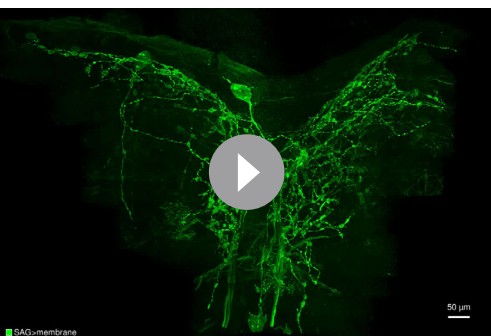

50 μm

☐ SAG>membrane

**Video 4.** Representative example of identifying SAG to pC1 connections via the SAG presynaptic site contact with the pC1 membrane. SAG and pC1 were labeled by membrane reporters. SAG and pC1 were semiautomatically segmented, and neuron masks were generated. Presynaptic sites labeled by BRP were automatically classified and assigned to SAG. Connections between SAG presynaptic sites and the pC1 neuron mask were automatically identified.
https://elifesciences.org/articles/81248/figures#video4

## Chemical connectivity between identified neurons

To test whether ExLLSM counts of chemical connections between identified neurons are also similar to counts obtained from EM, we focused on connections from the two cholinergic SAG neurons to their primary downstream targets, the 10 pC1 neurons (*Wang et al., 2020a*; *Figure 4A*). Most *Drosophila* synapses are polyadic (*Scheffer et al., 2020*), such that a single presynaptic site has multiple postsynaptic contacts. In the FIB-SEM hemibrain dataset, the two SAG neurons make 5534 connections to downstream neurons from 1198 presynaptic sites. Of these connections, 938 are made to pC1 neurons via 677 SAG presynaptic sites.

Although ExLLSM is in principle compatible with any primary antibody, there is no known postsynaptic analog to BRP that labels all or most postsynaptic sites in *Drosophila*. Additionally, genetic, histochemistry, or imaging constraints make it difficult to specifically label distinct molecules in different neuron types in the same animal. In particular, our current ExLLSM protocol is limited to three-color imaging, which makes it challenging to simultaneously visualize two cell types and two molecular markers. To circumvent this problem, we tested whether we could quantify connectivity in two ways: (1) by labeling all presynaptic sites, the membrane of presynaptic neurons, and postsynaptic sites only in the postsynaptic neurons (*Figure 2—figure supplement 3B*, *Figure 4B–F*); and (2) by labeling all presynaptic sites and the membranes of both pre- and postsynaptic neurons (*Figure 2—figure supplement 3C*, *Figure 4I–O*). The first approach allows us to quantify both the total number of SAG>pC1 connections (938 from EM) and the number of SAG presynaptic sites making these connections (677 from EM). The second approach only allows us to quantify the number of SAG presynaptic sites making connections to pC1 (677 from EM).

For both methods, we used the LexAGAD driver line to label the SAG neurons (*Feng et al., 2014*), a split-GAL4 line for the pC1 neurons (*Wang et al., 2020a*), and the BRP antibody to label all presynaptic sites (*Figure 4A*). In the first approach, we labeled putative cholinergic receptors in pC1 using a genetic reporter that labels Drep2 proteins in a neuron-specific manner. Although this reporter does not label endogenous Drep2 protein, it colocalizes with the acetylcholine receptor subunit Dα7 at cholinergic synapses (*Andlauer et al., 2014*). Using this approach (*Figure 4B–F*, *Video 3*), we counted an average of 729 SAG to pC1 connections made by an average of 583 presynaptic sites (n = 11) (*Figure 4G*). Both averages are slightly lower than the counts obtained from the single FIB-SEM hemibrain sample (938 and 677, respectively), but we note that the counts for this EM sample fall within the range we obtained by ExLLSM (*Figure 4G*).

Using the second approach to quantify connectivity, which quantifies the number of SAG presynaptic sites that contact the pC1 membrane (*Figure 4I–O*, *Video 4*), we obtain an average of 501 connections (n = 6) (*Figure 4G*). The most connections we counted in a single sample was 604, below the count of 677 in the EM sample, suggesting that this method may more consistently undercount synapses. Postsynaptic sites are often located in fine neural processes (*Schneider-Mizell et al., 2016*; *Scheffer et al., 2020*), and without specific labeling using a postsynaptic marker such as Drep2, these fine processes may be missed more often using ExLLSM than EM due to the discontinuities inherent in the immunohistochemistry and 8× expansion methods.

We conclude that the ExLLSM reconstruction strategies used here result in similar connection counts that are in general agreement with those obtained by EM. We recommend the use of a specific postsynaptic marker where practical to avoid undercounting. Nonetheless, we note that all of these approaches, including EM (*Scheffer et al., 2020*), rely on detection methods that generally have high

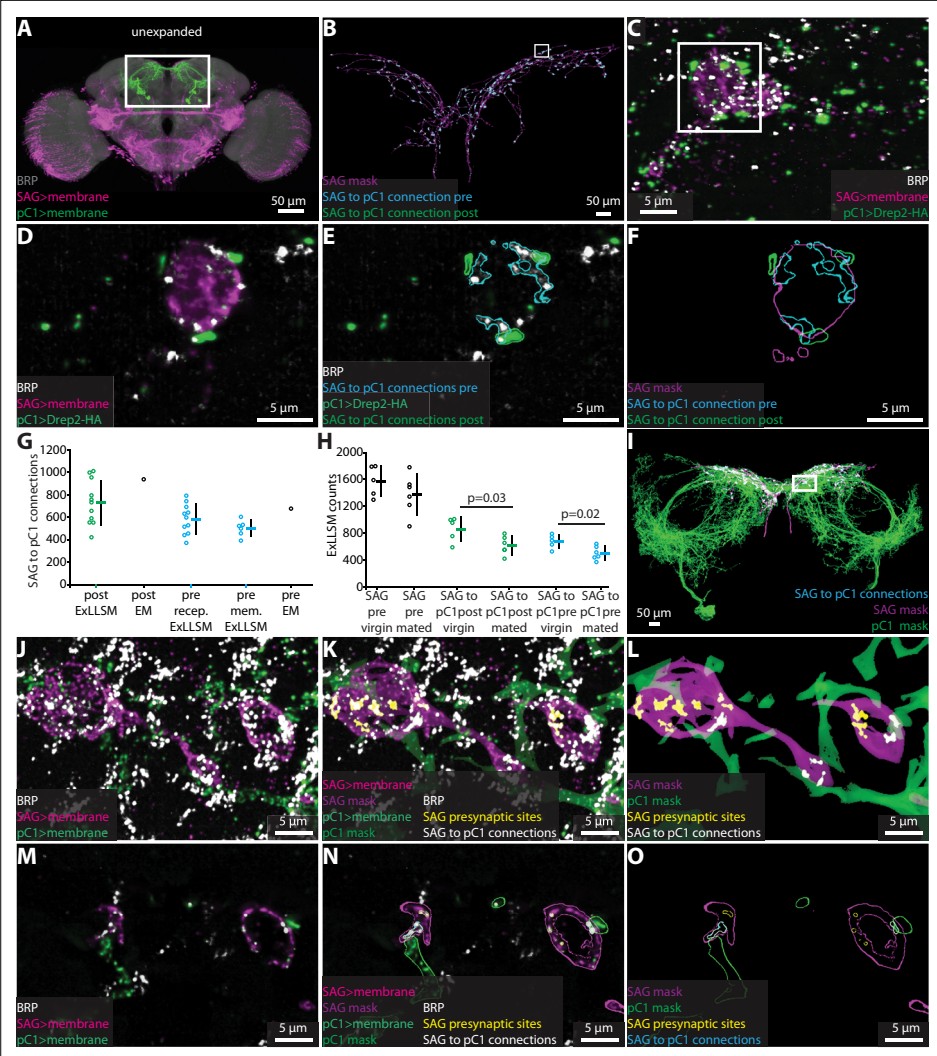

**Figure 4.** Quantifying connectivity using ExLLSM. (**A**) Unexpanded female brain with SAG (and off-target) neurons (magenta), pC1 (green) neurons, and BRP presynaptic sites (nc82, gray) labeled. (**B–F**) Representative examples of quantifying connectivity via pC1 postsynaptic site contact with SAG presynaptic sites. (**B**) Segmented SAG mask and the automatically detected pre- (cyan) and postsynaptic sites (green) of SAG to pC1 connections. (**C**) Max intensity projection of 200 z-slices from region approximately in the rectangle in (**B**). (**D–F**) Single z-slice of region in the rectangle from (**C**). (**G**) Quantification of SAG to pC1 connections. SAG to pC1 connections as quantified by the number of pC1 postsynaptic sites that contact SAG presynaptic sites in 11 ExLLSM samples (post ExLLSM) and the number of pC1 postsynaptic sites that connect to SAG presynaptic sites in one electron microscopy (EM) sample (connections, post EM). SAG presynaptic sites that make connections to pC1 as quantified by the number of SAG presynaptic sites that contact pC1 postsynaptic sites in 11 ExLLSM samples (pre recep. ExLLSM), number of SAG presynaptic sites that contact the pC1 membrane in six ExLLSM samples (pre mem. ExLLSM), and number of SAG presynaptic sites that make connections to pC1 postsynaptic sites in one EM sample (pre EM). (**H**) Quantification of SAG presynaptic sites and SAG to pC1 connections in virgin and mated female ExLLSM samples. Number of SAG presynaptic sites in five virgin samples (SAG pre virgin) and six mated samples (SAG pre mated). SAG to pC1 connections as quantified by the number of pC1 postsynaptic sites that contact SAG presynaptic sites in five virgin samples (SAG to pC1 post virgin) and in six mated samples (SAG to pC1 post mated). SAG presynaptic sites that make connections to pC1 as quantified by the number of SAG presynaptic sites that contact pC1 postsynaptic sites in five virgin samples (SAG to pC1 pre virgin) and six mated samples (SAG to pC1 pre mated). Virgin pre- and postsynaptic ExLLSM counts are significantly different from mated pre- and postsynaptic ExLLSM counts (*t*-tests). Individuals and mean ± SD plotted in (**G, H**). (**I–O**) Quantifying connectivity via pC1 membrane contact with SAG presynaptic sites. (**I**) Segmented masks of SAG and pC1 and SAG to pC1 connections approximately in rectangle in (**A**). (**J–L**) 100 z-slices of region approximately in the rectangle in (**I**). (**M–O**) Single z-slice of data from (**J–L**). Mask outlines shown.

specificity but lower sensitivity, and therefore all undercount connections to various degrees. For practical purposes, however, relative synaptic counts are usually more informative than absolute numbers, and so a slight but consistent undercounting is not necessarily problematic. ExLLSM further mitigates this concern because it enables a much larger number of samples to be surveyed than EM, providing a significantly more accurate assessment of relative connectivity regardless of which labeling strategy is used.

To assess the power of ExLLSM to reveal relative differences in synaptic connectivity, we tested the hypothesis that SAG neurons make more synaptic connections with pC1 neurons in virgin females than in mated females. The pC1 neurons regulate female receptivity and egg-laying (*Wang et al., 2020a*; *Wang et al., 2020b*), both of which change dramatically after mating. After mating, sensory neurons in the uterus detect the presence of a male seminal fluid protein (*Häsemeyer et al., 2009*; *Yang et al., 2009*), and the SAG neurons relay this signal from the uterus to pC1 neurons in the brain (*Feng et al., 2014*). Both the SAG and pC1 neurons have higher basal activity in virgin females than in mated females (*Feng et al., 2014*; *Wang et al., 2020a*), and so we hypothesized that they may also have more synaptic connections in virgin females than in mated females. We tested this hypothesis by counting SAG>pC1 synapses in ExLLSM brain samples from a total of five virgins and six mated females using the Drep2 labeling strategy. The average number of SAG presynaptic sites was similar in mated and virgin females (*Figure 4H* , *t*-test, two-tailed p=0.26) but 25% fewer of these presynaptic sites were connected to pC1 in mated females, resulting in 28% fewer connections (*Figure 4H*, *t*-test, two-tailed p=0.03 and 0.02, respectively). These data establish that SAG>pC1 synapses are indeed remodeled after mating and, moreover, demonstrate the power of ExLLSM to reveal and quantify state-dependent changes in neuronal connectivity.

## Detection and characterization of electrical connections

Neurons also communicate through electrical connections called gap junctions (*Güiza et al., 2018*; *Nagy et al., 2018*), which are difficult to detect in EM images. Invertebrate gap junction channels are formed by innexin proteins (*Syrjanen et al., 2021*), which provide a molecular label to visualize and quantify electrical connections using ExLLSM. Functional gap junctions are generally composed of two hemichannels, one from each neuron. Antibody staining against gap junction proteins reveals both distributed punctate signals and pronounced aggregations (*Phelan et al., 2008*; *Markert et al., 2016*). Punctate signals may label hemichannels, which themselves may be functional channels (*Skerrett and Williams, 2017*), whereas aggregations plausibly label gap junctions composed of two hemichannels.

In *Drosophila*, there are eight types of innexin proteins (*Stebbings et al., 2002*). We used an antibody against innexin 6 (INX6) to assess the ability of our ExLLSM pipeline to detect and classify possible electrical connections. As previously reported (*Wu et al., 2011*; *Troup et al., 2018*; *Ammer et al., 2022*), INX6 immunoreactivity was most pronounced in the fan-shaped body (*Figure 5A–D*). In expanded tissue, the fan-shaped body INX6 aggregations were revealed to be composed of distinct

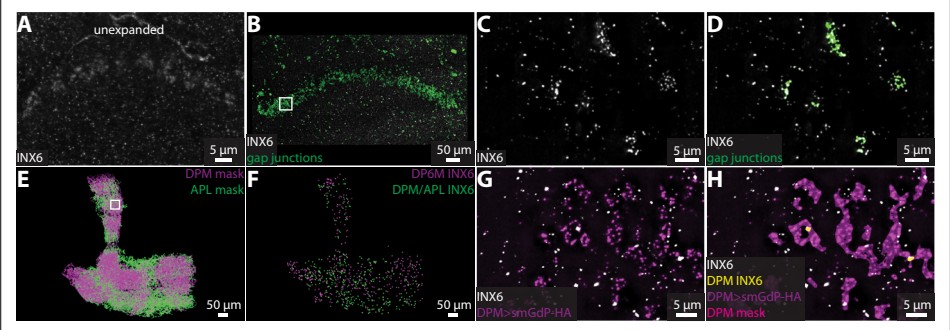

**Figure 5.** Visualizing gap junction proteins with ExLLSM. INX6 immunoreactivity in the fan-shaped body in the unexpanded (**A**) and expanded (**B**) brain. (**B**) INX6 and automatically detected putative gap junctions (green). (**C–, D**) Zoom in on region in rectangle in (**B**). (**E**) Segmented DPM and APL neuron masks. (**F**) Automatically detected possible DPM and APL gap junction sites. (**G, H**) Zoom in on region in rectangle from (**E**) with INX6 (white), the DPM membrane (purple), DPM mask (magenta), and possible DPM gap junction sites (yellow).

clusters of INX6 (*Figure 5C*). These INX6 clusters generally resembled the clusters of pre- and post-synaptic proteins found at chemical synapses (*Figures 2–4*) and could be automatically classified as possible gap junctions using the same workflows used to detect chemical synapses (*Figure 5B–D*).

We also visualized INX6 immunoreactivity in the mushroom body, where gap junctions essential for memory formation are formed between DPM and APL neurons (*Wu et al., 2011*). Using ExLLSM, we found that INX6 immunoreactivity was weak and punctate in the mushroom body, with few if any clusters (*Figure 5G and H*). These punctate signals colocalized with DPM and APL neurons (*Figure 5E-H*). We found that DPM-associated INX6 contacted APL membranes and APL-associated INX6 contacted DPM membranes, but did not detect any contact between INX6 in DPM and INX6 in APL. It is unclear whether the punctate INX6 detected in these neurons labels hemichannel portions of heterotypic gap junctions. Previous findings indicate that electrical coupling between DPM and APL neurons is mediated by heterotypic gap junctions composed of INX6 and INX7, respectively (*Wu et al., 2011*), which suggest that we may be visualizing part of the gap junctions formed between these neurons.

Although additional work is necessary to characterize the structure of hemichannels and homo- and heterotypic gap junctions in superresolution LM, these data demonstrate the ability of our ExLLSM pipeline to detect electrical as well as chemical connections. Using the recently developed suite of antibodies to label all eight *Drosophila* innexins (*Ammer et al., 2022*), ExLLSM has the potential to greatly accelerate the quantification of electrical connectivity between neurons and extend correlative structure–function studies to electrical as well as chemical connections.

## Linking circuit structure to neurophysiology and behavior across individuals

To assess the potential of our ExLLSM pipeline to perform correlative studies of circuit structure, physiology, and function, we examined chemical connectivity within the neural circuit for male courtship song in *Drosophila*. Male song is a highly stereotyped yet variable behavior that can be readily quantified, and for which critical nodes in the underlying circuitry have been identified.

We focused our analysis on two neuron types in the male brain that function in song production, pC2l and pIP10 (*Figure 6A*). The male courtship song consists of a series of loud pulses, interspersed with bouts of continuous humming known as the sine song (*Greenspan and Ferveur, 2000*; *Figure 6B and C*). The pC2l neurons respond to both auditory and visual cues (*Kohatsu and Yamamoto, 2015*; *Deutsch et al., 2019*), and activating a subset of these neurons elicits pulse song acutely, followed by sine song post-activation (*Deutsch et al., 2019*). Here, using a different genetic driver line that labels eight pC2l neurons in each hemisphere, we found that pC2l activation produced acute courtship song in most flies with natural pulse and sine characteristics (*Figure 6B*, *Figure 6—figure supplement 1A and B*). pIP10 cells are bilaterally paired male-specific descending neurons. pIP10 activity is necessary and sufficient for the production of pulse song and, to a lesser extent, sine song, and influences song choice during courtship (*von Philipsborn et al., 2011*; *Clemens et al., 2018*; *Calhoun et al., 2019*; *Ding et al., 2019*).

To examine individual variability in the structure and function of the pC2l-pIP10 circuit, we expressed the red-light activated cation channel CsChrimson (*Klapoetke et al., 2014*) in pC2l neurons using a split-LexA line and the calcium indicator GCaMP6f (*Chen et al., 2013*) in pIP10 neurons using a split-GAL4 line (*Ding et al., 2019*; *Figure 6A*). In seven individual flies, we optogenetically activated pC2l and quantified the song generated (*Figure 6*), then paired a male with a female for 10 min and quantified the courtship song produced upon natural stimulation (*Figure 6C*). We then removed the brain, optogenetically activated pC2l. and measured the calcium response in pIP10 (*Figure 6D*). Finally, we used ExLLSM to quantify structural connectivity between pC2l and pIP10 in each brain (*Figure 6E and F*). With this protocol, we could assess how courtship behavior, optogenetically elicited behavior, functional connectivity, and structural connectivity are all related to each other across individual flies (*Figure 6G and H*, *Figure 6—figure supplement 1C and D*).

We established that pC2l cells are presynaptic to pIP10, counting an average of 92 pC2l>pIP10 connections per brain. In these experiments, we had to rely on GCaMP6 as the pIP10 label rather than the membrane marker used in the previous experiments with SAG and pC1 cells (*Figure 4*), which likely further but consistently undercounts the number of connections. There was a strong linear relationship between the number of synaptic contacts and the strength of the functional connection between pC2l and pIP10 cells across animals (Pearson's $r$ = 0.88, p=0.009, *Figure 6G*). Structural

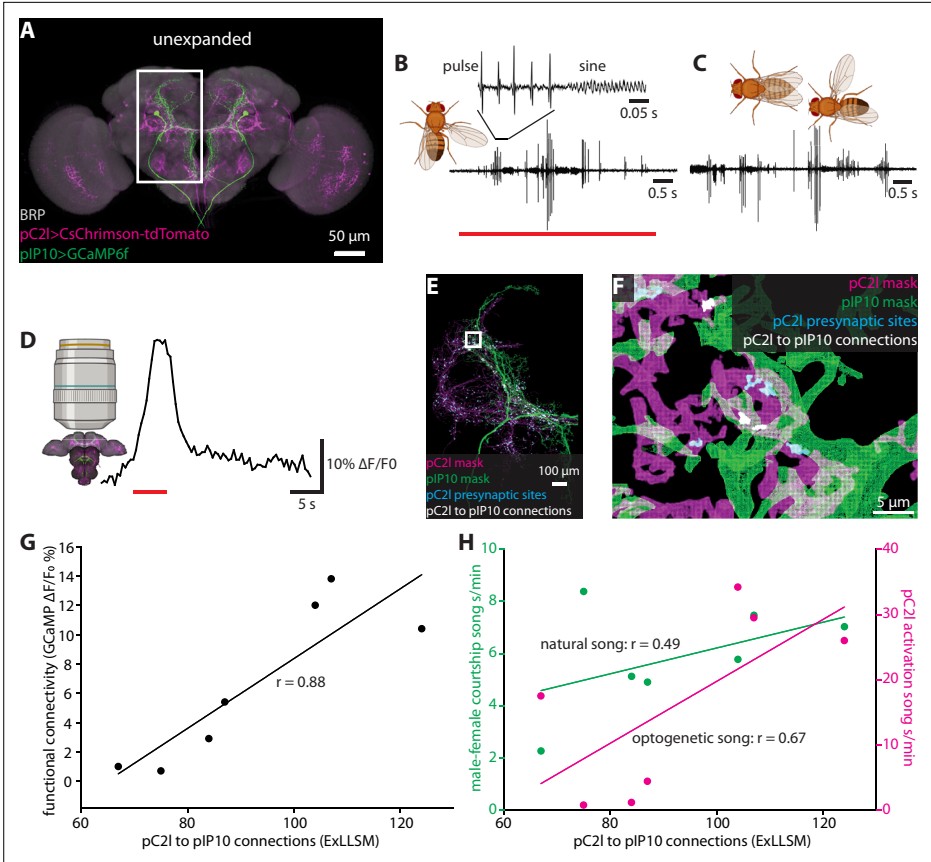

**Figure 6.** Correlating individual variability in structural connectivity, functional connectivity, and behavior. (**A**) Unexpanded brain of a male fly with presynaptic sites (BRP, gray), GCaMP6f (green) in pIP10, and CsChrimson (magenta) in pC2l(and off-target neurons) labeled (**B**) Representative example of optogenetic activation of pC2l (red bar), resulting in the acute production of pulse and sine song. (**C**) Representative example of song produced by the male during a courtship bout with a female. (**D**) Representative example of a GCaMP6f response (black trace) in pIP10 induced by optogenetic activation of pC2l (red bar). (**E**) Representative 8× view of a segmented pC2l, segmented pIP10, pC2l presynaptic sites, and the pC2l to pIP10 connections as measured by pIP10 contact with pC2l presynaptic sites. (**F**) Zoom in on region in the rectangle from (**E**). (**G, H**) ExLLSM connections plotted against (**G**) the strength of the functional connection between pC2l and pIP10 (as in **D**), and (**H**) song seconds/ minute elicited by optogenetic pC2l activation (magenta, as in **B**) and by a female fly (green, as in **C**). Linear regression and associated Pearson's correlation coefficient (r) plotted for each relationship.

The online version of this article includes the following figure supplement(s) for figure 6:

**Figure supplement 1.** Quantification of courtship song features, correlating individual variability in optogenetically induced pulse and sine song with structural connectivity, correlating individual variability in song with functional connectivity, and correlating individual variability in GCaMP fluorescence with structural connectivity.

connectivity between pC2l and pIP10 was less strongly correlated with optogenetically elicited song (Pearson's *r* = 0.67, p=0.09, *Figure 6H*), and least of all with naturally stimulated song (Pearson's *r* = 0.49, p=0.27, *Figure 6H*).

We also observed a strong linear correlation between the functional connection and optogenetic song (Pearson's *r* = 0.83, p=0.02, *Figure 6—figure supplement 1D*), more so than between structural connectivity and optogenetic song (Pearson's *r* = 0.67, p=0.09, *Figure 6G*) and between functional connectivity and naturally stimulated song (Pearson's *r* = 0.37, p=0.41, *Figure 6—figure supplement 1D*). These relationships are unlikely to merely reflect individual variations in expression levels of the GCaMP6 reporter as there was no correlation between the baseline fluorescence of the GCaMP signal and the structural connection (Pearson's *r* = −0.09, p=0.85, *Figure 6—figure supplement 1E*).

Previous evidence suggests that pC2l and pIP10 directly drive the pulse component of courtship song, with only indirect contributions to sine song production (*Deutsch et al., 2019*; *Ding et al., 2019*; *Roemschied et al., 2021*). If so, one might expect pC2l>pIP10 connectivity to have a stronger relationship with pulse song than with either sine or total song. Indeed, separating song into its pulse and sine components revealed that structural connectivity was more strongly correlated with optogenetic pulse song production (Pearson's $r$ = 0.75, p=0.054, *Figure 6—figure supplement 1C*) than total song (Pearson's $r$ = 0.67, p=0.09, *Figure 6H*) or sine production alone (Pearson's $r$ = 0.11, p=0.82, *Figure 6—figure supplement 1C*). These data demonstrate that ExLLSM allows a variable feature of behavior to be correlated with individual differences in structural connectivity.

## Discussion

Our goal in this project was to develop an ExLLSM pipeline for the rapid and targeted reconstruction of neural circuits. We developed and tested this pipeline using *Drosophila* as a model due to the availability in this species of reference connectomes (*Takemura et al., 2015*; *Takemura et al., 2017*; *Bates et al., 2020*; *Scheffer et al., 2020*) and genetic tools for highlighting cells and molecules of interest (*Jenett et al., 2012*; *Andlauer et al., 2014*; *Chen et al., 2014b*; *Tirian and Dickson, 2017*; *Dionne et al., 2018*). The protocols we have developed allow a single sample to be prepared and analyzed within a week and can be readily multiplexed to process several samples in parallel. Although LLSM provides an excellent combination of imaging resolution and speed with limited photobleaching (*Gao et al., 2019*), other light sheet microscopes can also be utilized to image large volumes of 8× expanded samples. As such, our method relies on relatively affordable microscopy and computational resources that are widely available, bringing connectomics research within reach of smaller labs. In establishing this pipeline, we developed new software tools to visualize and analyze the high-resolution multi-terabyte datasets generated. These tools are freely available (*Lillvis et al., 2021*) and have been designed with efficiency and flexibility in mind. Accordingly, these tools should be well suited for any organism in which neurons can be reliably labeled and tissue can be sufficiently expanded. This includes other *Drosophila* species, nematodes, rodents, zebrafish, and organoids, for each of which genetic tools to label specific neurons are available and expansion microscopy protocols have been established (*Freifeld et al., 2017*; *Gao et al., 2019*; *Yu et al., 2020*; *Rodriguez-Gatica et al., 2022*). The development of such genetic tools and expansion protocols in other species will extend the applicability of this approach across taxa. Furthermore, the convolutional neural network models for synapse and neuron segmentation are classifiers of high signal punctate and continuous structures, respectively. As such, the models may already work well for segmenting similar structures from other species or microscopes. If not, these models can be retrained with a suitable ground truth data set and the entire computational pipeline applied to these new systems.

We anticipate that many further developments will be spurred by the power and low entry cost of ExLLSM-based circuit analysis. With the development of neuron labeling methods that leave fewer gaps in the fluorescent signals of expanded samples, automated single-neuron segmentation should become feasible (*Januszewski et al., 2018*), allowing connectivity to be assessed at the level of single cells rather than single-cell types, as we have done here. Such methods would likely also allow more densely labeled neurons to be traced, ultimately perhaps even entire connectomes (*Gao et al., 2019*). A further improvement would come through the generation of reagents to visualize specific components of chemical and electrical synapses, ideally in the form of genetic tools that label endogenous molecules in a cell-type-restricted manner (*Chen et al., 2014b*). Because ExLLSM enables correlative structural and functional studies, probes that reveal the functional state as well as the location of such molecules would be particularly valuable.

We have shown here that comparative ExLLSM can be used to reveal state-dependent differences in neuronal connectivity. More generally, the method is ideally suited to explore how genetic, environmental, and stochastic processes work together to create individual differences in neuronal connectivity. In combination with new methods to genetically identify and label homologous neurons across species (*Stern et al., 2017*; *Tanaka et al., 2017*; *Seeholzer et al., 2018*; *Ding et al., 2019*; *Auer et al., 2020*), this approach will also enable new studies on circuit and behavior evolution. By facilitating the collection of both functional and structural data from the same samples, our ExLLSM pipeline also allows individual and species differences in circuit structure to be correlated with differences in neurophysiology and behavior. Our ExLLSM approach to neural circuit reconstruction thereby fills a

critical methodological gap in exploring the links between genes, structure, physiology, and behavior, and should be a powerful tool in efforts to understand how connectomes work and how they evolve.

## Materials and methods

### Experimental animals

Unless noted otherwise, flies were raised on standard cornmeal-agar-based medium on a 12:12 light/dark cycle at 25°C. Detailed information on fly genotypes, sex, housing, and age for each experiment is indicated in the relevant section below and in *Supplementary file 1*.

### Genetic reagents

LexA, split-GAL4, and split-LexA lines used in this study have constructs inserted at the attP40 or attP2 landing sites (*Tirian and Dickson, 2017*; *Dionne et al., 2018*), unless noted otherwise (*Supplementary file 1*). Unpublished LexA, p65ADZp, ZpGAL4DBD, and ZpLexADBD lines labeling neurons of interest were identified using a color depth maximum intensity projection mask search (*Otsuna et al., 2018*). The expression of driver lines was examined with a UAS or LexAop reporter by immunofluorescence staining and confocal microscopy (https://www.janelia.org/project-team/flylight/protocols). For split lines, the combinations of p65ADZp and ZpGAL4DBD or p65ADZp and ZpLexADBD that gave the most specific expression patterns were stabilized by putting the two hemi-drivers in the same flies, and SS and SL (denoting stable split-GAL4 or split-LexA, respectively) numbers were assigned. SS and SL combinations were checked for expression in the same fly to ensure that no off-target neuron expression was found in the overlapping regions of interest due to unintended interactions between SS and SL p65ADZp, ZpGAL4DBD, ZpLexADBD components.

### Immunohistochemistry

For ExLLSM experiments, brains were dissected in external saline composed of (in mM) 103 NaCl, 3 KCl, 5 N-tris(hydroxymethyl) methyl-2-aminoethane-sulfonic acid, 10 trehalose dihydrate, 10 glucose, 26 NaHCO$_3$, 1 NaH$_2$PO$_4$, 4 MgCl$_2$, 3 KCl, 2 sucrose, and 1.5 CaCl$_2$ (280–290 mOsm, pH 7.3; all components from Sigma). All subsequent washes and incubations were conducted on a rocker or rotator. After dissection, samples were fixed in 2% formaldehyde (Electron Microscopy Solutions, 20% stock diluted in external saline) at room temperature for 55 min, washed in three 20 min PBST (0.5% Triton X-100 in 1× phosphate-buffered saline) washes, and blocked using 5% normal goat serum (diluted in PBST) for 90 min. After blocking, samples were incubated in primary antibodies at 4°C for 2–3 days, washed for 2–5 total hours at room temperature in at least five PBST washes, and then incubated in secondary antibodies at 4°C for 2 days. All antibodies and concentrations are listed in *Supplementary file 1*. Finally, samples were washed for 2–5 total hours at room temperature in at least five PBST washes and stored in 1× phosphate-buffered saline at 4°C until they were prepared for expansion, which occurred within 24 hr. Unexpanded images shown were prepared by the Janelia Fly Light Project Team. Unexpanded samples were prepared in a largely similar manner to expanded samples and mounted in DPX on a glass slide. For detailed unexpanded brain dissection, immunohistochemistry, and DPX mounting protocols, see https://www.janelia.org/project-team/flylight/protocols.

### 8× expansion

Acryloyl-X, SE (6-((acryloyl)amino)hexanoic acid, succinimidyl ester; here abbreviated AcX; Thermo Fisher) was resuspended in anhydrous DMSO at a concentration of 10 mg/mL, aliquoted, and stored frozen in a desiccated environment. AcX stock solution was diluted in 1× PBS to a final concentration of 0.1 mg/mL AcX. Specimens were incubated in this 0.1 mg/mL AcX solution overnight at room temperature. Gelation chambers were created by adhering silicone gaskets (e.g., Sigma, GBL665504) to poly-lysine-treated glass slides. Specimens were immobilized on the poly-lysine-treated surface, at least 2 mm away from the silicone surface.

A 4 M sodium acrylate stock solution was prepared by combining 5.5 mL acrylic acid (Sigma, 147230), 4.5 mL water, and 7.2 mL 10 M NaOH in a fume hood, then adding 1 M NaOH until the pH reached 7.75–8, and finally adding water up to 20 mL. This solution should be clear. Monomer solution (1× PBS, 1 M NaCl, 1.84 M sodium acrylate, 2.5% acrylamide, 0.05% N,N′-methylenebisacrylamide) was mixed, frozen in aliquots, thawed fully, vortexed, and cooled on ice before use.

Concentrated stocks of the initiator ammonium persulfate (APS, 10 wt%), accelerator tetramethy-lethylenediamine (TEMED, 10 vol%), and inhibitor 4-hydroxy-2,2,6,6- tetramethylpiperidin-1-oxyl (4-HT, 0.5 wt%) were prepared as concentrated stock solutions, which were frozen in aliquots and then fully thawed and vortexed before use. Initiator, accelerator, and inhibitor stock solutions were added to the monomer solution at a ratio of 2 µL each per 94 µL monomer solution to produce gelation solution. AcX-anchored specimens were washed 3 × 10 min in gelation solution, on ice. Gelation chambers were sealed with a 22-mm-square coverslip, excess gelation solution was removed, and the sealed chambers were transferred to 37°C for 2 hr, protected from light, for gelation.

Proteinase K (New England Biolabs) was diluted 1:100 to 8 units/mL in digestion buffer (50 mM Tris [pH 8], 1 mM EDTA, 0.5% Triton X-100, 500 mM NaCl) to produce proteinase solution. Gels were recovered from chambers, trimmed close to the specimens, and incubated fully immersed in proteinase solution overnight at room temperature, with shaking. Gels were washed in 1× PBS for 30 min to remove proteinase K. Digested gels were next incubated on ice, with shaking, in at least a tenfold excess volume of gelation solution 2 × 10 min, with APS omitted in the first two incubations. A 5 mL syringe filled with silicone grease was used to apply four dabs of grease per specimen to glass slides, at the corners of one ~20 mm square per specimen. Each gel was transferred to the middle of one of these squares and covered with a 22-mm-square coverslip. This coverslip was pressed down to gently but fully contact the gel, while being held in place by the dabs of vacuum grease. Excess gela-tion solution was backfilled into the resulting chamber to impede access of atmospheric oxygen to the gel. The completed chamber was moved to the 37°C incubator for 2 hr for gelation. The resulting doubly gelled specimen was recovered from the chamber, and excess gel was trimmed away, followed by staining in 0.2 mg/mL DAPI in 1× PBS for 2 hr. The gel was incubated in ~50 mL of doubly deion-ized water for 12–24 hr to expand.

## Lattice light sheet imaging

Sample mounting was highly similar to Gao et al. 2019 with some slight modifications. A 12-mm round-glass coverslip was brushed with a solution composed of poly-L-lysine hydrobromide (0.078% weight/ volume; Sigma-Aldrich: P1524) and Kodak Photo-Flo 200 (0.2% volume/volume; Electron Microscopy Sciences: 74257) and allowed to dry. Using a razor blade, 8× expanded gels were trimmed close to the tissue sample in x, y, and z by viewing DAPI staining under a wide-field microscope (Olympus MVX10). Liquid around the gel was wicked away with a Kimwipe (Kimtech Science) and the gel was transferred onto the dry, coated coverslip attached to a sample holder by metal clamps. The sample holder was attached to the LLSM sample chamber that was filled with 1 mM Tris base, which was used to reduce sample shrinking (possibly due to acidification from atmospheric $CO_2$) during long imaging runs.

The imaging region of interest (ROI) was identified by taking a snapshot of the neuron fluorescence using a wide-field camera mounted under the sample bath. An ROI was drawn and the x, y coordi-nates of the ROI relative to the stage position were calculated. The z coordinates were determined by scanning through the sample and identifying the upper and lower bounds of the ROI. The z-offset for each wavelength, which sets the precise position of the light sheet relative to the detection plane of the objective, was determined by taking small z-stacks of each channel, reslicing the stack, and creating a MIP of the resliced stack to view the symmetry of the XZ PSF. Offset was adjusted until the PSF was symmetric, similar to the procedure described in Figure 5 of *Gao et al., 2019*. Adjustments of up to 1 µm are common to account for the slight mismatch between the refractive index of the gel and buffer.

LLSM hardware and software setup and control were near identical to the 'LLSM optimized for expanded samples' described in *Gao et al., 2019*, with some slight modifications described here. Samples were excited by 488, 560, and 642 nm lasers. Emission was split between two detection cameras using a 561 nm dichroic mirror (Semrock: Di03-R561-t1−25x36). In front of the green camera, FF02-525/40-25, FF01-432/515/595/730-25, and NF03-405/488/561/635E-25 emis-sion filters (Semrock) were used. In front of the red camera, FF01-440/521/607/700-25 and NF03-405/488/561/635E-25 emission filters (Semrock) were used. All samples were imaged in objective scan mode. Imaging tile sizes ranged from 360 × 992 × 501 pixels to 360 × 1600 × 501 pixels. We achieved a final resolution of 30 × 30 × 100 nm, as calculated from the FWHM of the 560 nm PSF in

XYZ. This is slightly lower than the best possible resolution for the beam NA (0.517–0.55); the light sheet was tuned to be slightly thicker to allow for sample variation and small amounts of instrument drift.

For large samples with non-rectangular processes, imaging tiles with no signals were automatically avoided by the software using the 'intelligent tiling' technique described in *Gao et al., 2019*. This strategy significantly reduced imaging time compared to standard tiling while capturing the signal of interest. The sample bath remained static throughout imaging, but imaging was occasionally paused to refill the bath to ensure the sample remained submerged during long imaging runs.

### Image processing and analysis overview

The goal of image analysis here was to quantify synaptic connectivity between neuron types (e.g., neuron 1 and neuron 2) labeled by transgenic fly lines. To accomplish this, image tiles were processed and stitched. Then fluorescently labeled neurons, presynaptic, and, in some cases, postsynaptic sites were segmented. Next, a colocalization analysis assigned the classified pre- and/or postsynaptic sites to segmented neuron masks. Finally, an additional colocalization analysis identified connections between the neurons by finding presynaptic sites from neuron 1 that contact neuron 2 or postsynaptic sites from neuron 2 that contact neuron 1 presynaptic sites (*Figure 2—figure supplement 3*).

To maximize accessibility and portability of our processing and analysis tools, we are distributing executable Docker containers (*Merkel, 2014*) for all of the code, making it easy to run the code across a wide range of systems through the use of Singularity (*Kurtzer et al., 2017*). The use of Nextflow further reinforces that goal by allowing our computational workflows to execute on any compute cluster or cloud, including but not limited to IBM Platform LSF, SLURM, and AWS Batch. By assembling the workflows into Nextflow pipelines, we also minimize the dependencies that are necessary for the user to install and provide a consistent command-line interface for invoking workflows and specifying runtime options. Each step of the pipeline is described below. Code and additional documentation to run all steps of the analyses described can be found at https://github.com/JaneliaSciComp/exllsm-circuit-reconstruction (*Lillvis et al., 2021*; *Lillvis, 2021*; *Rokicki and Lillvis, 2021*; *Rokicki and Kawase, 2021*).

### Image preprocessing and stitching

Image preprocessing (flat-field correction and deconvolution), stitching, and stitched N5 (https://github.com/saalfeldlab/n5) and TIFF series exports were conducted as in *Gao et al., 2019*. Default preprocessing and stitching parameters were also identical to those listed in *Gao et al., 2019*. We include all of these published preprocessing and stitching steps (https://github.com/saalfeldlab/stitching-spark) in our analysis pipeline (https://github.com/JaneliaSciComp/exllsm-circuit-reconstruction). The entire preprocessing and stitching pipeline can be run as a workflow with flexible parameter setting options, or each step can be run independently.

### Visualization

For 3D visualization, preliminary image analysis, user guided semiautomatic neuron segmentation, ground truth data generation, and video creation, we extended the free and open-source VVD Viewer software (https://github.com/JaneliaSciComp/VVDViewer; *Wan et al., 2012*) to handle high-resolution big image (data sizes ≤5 TB/channel tested) LM datasets like those generated by ExLLSM. N5 directories generated by the stitching pipeline can be opened directly in VVD Viewer. For image analysis and segmentation, we recommend using multiscale pyramid VVD Viewer files that are more efficiently transferred across the network and faster to load into GPU memory. We include workflows to convert N5 or TIFF series to VVD Viewer pyramid files that can be run locally or on a compute cluster. A detailed manual for running VVD Viewer, creating VVD Viewer files, and analyzing ExLLSM data in VVD Viewer can be found at https://github.com/JaneliaSciComp/exllsm-circuit-reconstruction (*Lillvis et al., 2021*).

Small image crops were used to generate ground truth data for training the synapse classifier (see 'Automatic synapse classification'). The Fiji plugin 'N5-Viewer' was utilized to open stitched N5 directories and make these crops (https://github.com/saalfeldlab/n5-viewer).

## Semiautomatic neuron segmentation

The first task to accomplish in order to quantify synaptic connectivity between neurons was to segment the neurons of interest in the fluorescently labeled neuron images. The primary barrier to this was the large size of the image data. Manually segmenting neurons in the multi-terabyte high-resolution images is a very slow process, and existing methods and software to segment 3D LM images did not transfer well to the large ExLLSM datasets. Using clean genetic fly line reagents – where no off-target neurons are labeled in the ROI – may allow for basic image processing strategies (i.e., thresholding and size filtering) to be utilized for at least preliminary segmentation. However, this was not sufficient in most cases, and more sophisticated image segmentation strategies needed to be employed.

Therefore, we extended the VVD Viewer software (https://github.com/JaneliaSciComp/VVDViewer; *Wan et al., 2012*) to semiautomatically segment multi-terabyte ExLLSM datasets. See https://github.com/JaneliaSciComp/exllsm-circuit-reconstruction for a detailed manual on segmenting neurons in ExLLSM datasets using VVD Viewer. Briefly, we used the VVD Viewer Component Analyzer tool to automatically and rapidly (seconds) segment downsampled ExLLSM neuron signals from background signal based on a combination of voxel intensity and connected component size thresholds. This tool individually segments disconnected components (neurons, synapse, etc.), but does not separate components if they are connected or touching. Therefore, this tool works well to automatically segment images that label individual neurons, multiple disconnected neurons, or multiple connected neurons that are being analyzed as a unit (e.g., a class of neurons; e.g., DA1-lPN in *Figures 2B–D and 3K–P*). As such, analyzing the DA1-lPN data, for example, required relatively little human time. The semiautomatic neuron segmentation steps required a maximum of 1 hr per sample and all other steps were automated.

However, to segment individual neurons that contact each other, the Component Analyzer tool can be used for initial segmentation from background, but the user will need to manually segment individual neurons (e.g., *Figures 2D and 3B, E–J*). The difficulty of such manual segmentation can vary substantially depending on labeling density and signal quality. For instance, manually segmenting individual L2 outputs (*Figure 3*) took ~10 min/neuron, whereas segmenting a pair of SAG neurons from off-target neurons (*Figure 4*) took 1–5 hr depending on the sample. Of course, more densely labeled samples will take more time. Finally, while it is possible to segment individual neurons from entangled bundles as shown here and elsewhere (*Gao et al., 2019*), the expansion factor will need to be increased by an order or magnitude or more and neuron labels must be continuous to approach EM levels of reconstruction density.

The segmentation result generated by Component Analyzer was manually edited in VVD Viewer until the final mask appropriately segmented the neuron(s) of interest. This final segmented mask was saved as a TIFF series that retains the original voxel size (104 × 104 × 180 nm) and intensity values (*Figure 2—figure supplement 1*).

In order to allow fast segmentation from background and 3D editing of the large ExLLSM datasets, Component Analyzer runs on a downsampled VVD pyramid. Consequently, the segmented neuron TIFF series will generally over mask the neuron on its edges (*Figure 2—figure supplement 1C, F, I, and L*). We found that the best way to correct this was to apply a pixel intensity threshold to the TIFF series. Thresholding levels were determined by generating a MIP of the segmented mask TIFF series. In most cases, the thresholding value generated by *Huang and Wang, 1995* or Li (*Li and Tam, 1998*) method removed the edge over masking and generated a mask that was true to the neuron signal (*Figure 2—figure supplement 1C, F, I, and L*). However, in some cases, these values were too low and a manually determined pixel intensity threshold value that accurately masked the neurons was used.

At this stage, we have generated a mask that is true to the fluorescent signal of the neuron. However, at 8×, the fluorescent signal along neurons is not completely continuous due largely to gaps in fluorophore or tag expression and/or antibody labeling along the neuron (*Gao et al., 2019*; *Figure 2—figure supplement 1D, G, J, and M*). These gaps in signal were filled using a flexible, 3D component connecting algorithm. We connected gaps of 20 voxels or less, and iterated this process four times. This process reliably connected disconnected neuron components that were clearly part of a continuous neuron with minimal unwanted connections (*Figure 2—figure supplement 1D, G, J, and M*).

Finally, the components of this mask were analyzed, disconnected pixels separated by one pixel were connected, and remaining components smaller than 2000 voxels were removed (*Figure 2—figure supplement 1D, G, J, and M*). The result of these steps creates a binary mask of the neuron signal in the imaging volume. We include all of the post-VVD Viewer mask processing steps (MIP creation, voxel intensity thresholding, 3D gap filling, connected component analysis, voxel shape changing, and size filtering) in our ExLLSM analysis pipeline tools. Each of these steps can be run independently, and therefore utilized for other image processing needs, or as a single workflow with flexible parameter inputs. See https://github.com/JaneliaSciComp/exllsm-circuit-reconstruction for more details, usage examples, and tutorials.

## Automatic neuron segmentation

Although the semiautomatic segmentation method described above is relatively fast and can be done with little manual intervention, we sought to determine whether we could accomplish this task automatically. To do this, we generated a 3D U-Net convolutional neural network (*Çiçek et al., 2016*) to automatically segment neural signals from nonspecific antibody labeling and noise. For the purposes of this article, we are focusing on segmenting all neurons labeled as a group as opposed to segmenting individual neurons from each other. In order to secondarily segment individual neurons from each other, VVD Viewer can be utilized to do so manually.

Ground truth data was generated via the semiautomatic neuron segmentation process described above. Two 1024 × 1024 × 512 pixel crops were made from 17 samples (representing seven different neuron classes) for training. We used these data crops to train the U-Net for 150 epochs until the loss, accuracy, and error rates plateaued.

We evaluated this network by running it on full image volumes in five brain samples that were not included in the training. The output of the U-Net is a probability array with pixel values between 0 and 1. The entire automatic neuron segmentation pipeline included post-U-Net pixel intensity thresholding (here, a 0.8 threshold was used) and size filtering to remove components smaller than 2000 pixels. We compared the results of the entire automatic neuron segmentation pipeline to semiautomatically segmented ground truth data of these datasets. Because most of the pixels contain no neural signals, 1024 × 1024 × 500 pixel crops were made in regions where ground truth data was present. On these crops, the average precision was 95% and average recall was 79% (*Figure 2—figure supplement 2*).

Notably, off-target neurons present in samples were included in the results of the automatic neuron segmentation pipeline. In some instances, this would not affect connectivity analyses because there is no overlap between these off-target neurons and the pre- or postsynaptic neurons of interest. In many other instances, these off-target neurons would need to be removed before further analyses. Therefore, we elected to use the semiautomatic neuron segmentation via VVD Viewer strategy for all data analysis here.

However, this automatic approach was fast (2 TB 15,000 × 8000 × 10,000 pixel image volumes were segmented in just 10 min on the Janelia compute cluster) and worked relatively well despite limited training. Therefore, developing this approach for future work is likely to improve analysis efficiency. We include all code for training and evaluating the neuron segmentation U-Net and the trained model used here (https://github.com/JaneliaSciComp/exllsm-neuron-segmentation; *Lillvis, 2021*). Additional details, instructions, and workflows for running automatic neuron segmentation can be found at https://github.com/JaneliaSciComp/exllsm-circuit-reconstruction.

## ROI cropping

Even when using intelligent tiling to reduce the acquisition of image tiles without neuron signals, it was often the case that a significant image volume was present outside of the neurons of interest. Therefore, after segmenting the key neuron(s), a 3D ROI was identified by stepping through VVD mask TIFF series in z in Fiji (*Schindelin et al., 2012*). The neuron masks and stitched TIFF series were then cropped based on a 2D ROI generated in Fiji, and the first and last slice of the 3D ROI. All subsequent steps (VVD mask postprocessing and connectivity analyses) were conducted on the subvolumes generated after cropping. Analyzing these subvolumes significantly reduced computing time and expense. We include code to accelerate this process that can be run locally or submitted to a compute cluster (https://github.com/JaneliaSciComp/exllsm-circuit-reconstruction).

## Automatic synapse classification

Presynaptic sites can be identified as clusters of BRP proteins (*Ehmann et al., 2017*). Using 8× ExLLSM and labeling BRP with the nc82 antibody (*Wagh et al., 2006*) or the STaR-BRP reporter (*Chen et al., 2014b*), discrete clusters of fluorescent antibodies were present that, as expected (*Schneider-Mizell et al., 2016*), varied significantly in shape and size across the *Drosophila* brain (*Figure 2H-K*). We tested using ilastik (*Sommer et al., 2011*), a 3D VGG-shaped neural network (*Simonyan and Zisserman, 2014*), and 3D U-Net-shaped neural network (*Çiçek et al., 2016*) to segment these heterogeneous structures from nonspecific antibody labels and background signals. On our data, we found that the neural networks performed better than ilastik and similarly to each other, and that the U-Net was faster than the VGG. Therefore, we elected to train a U-Net convolutional neural network to automatically classify presynaptic sites.

To generate ground truth data for training the U-Net, we made 100 × 100 × 100 and 500 × 500 × 500 pixel crops of BRP staining (as labeled using the nc82 antibody) using the Fiji N5 Viewer. Based on the molecular architecture of BRP complexes (*Fouquet et al., 2009*; *Andlauer et al., 2014*), we considered clusters of three or more BRP labels in close proximity that fell along a common plane to be presynaptic sites. We semiautomatically segmented these presynaptic sites from unclustered antibody labels and background signals using VVD Viewer. This semiautomatic segmentation was accomplished similarly to semiautomatic neuron segmentation: the VVD Viewer Component Analyzer tool was used to extract signal from background followed by manual inspection of each potential presynaptic site. In total, we segmented over 10,000 presynaptic sites in image crops from 25 different brains. Crops were made from the optic lobe, mushroom body, lateral horn, central complex, antennal lobe, and protocerebrum. Ground truth data used to train the synapse classifier is available at Dryad (https://doi.org/10.5061/dryad.5hqbzkh8b).

We used these raw image data crops and manually segmented presynaptic sites to train the U-Net for 3000 epochs until the loss, accuracy, and error rates plateaued. The entire synapse classification and assignment pipeline includes a post-U-Net processing workflow. This post-U-Net workflow includes a watershed segmentation step to segment individual synaptic sites and a size filter to remove components below a given size threshold. Unclustered antibody signals were less than 400 pixels in size, and the vast majority of presynaptic sites were greater than 400 pixels. Therefore, and consistent with previous work, classified presynaptic sites that were smaller than 400 pixels were removed (*Gao et al., 2019*).

We evaluated the results of this synapse detection pipeline (including post-U-Net watershed segmentation and 400 pixel size thresholding) by running it on data crops of BRP labeled by nc82 from the optic lobe, protocerebrum, lateral horn, and mushroom body of five brain samples that were not included in the training. We compared these results to the manually segmented ground truth data (2300 presynaptic sites) of these image volumes. The final synapse detection pipeline had an average precision of 94% and recall of 88% (*Figure 2L-Q*).

In addition to labeling presynaptic sites by visualizing BRP, we labeled putative cholinergic postsynaptic sites by visualizing Drep2 (*Andlauer et al., 2014*) in pC1 neurons using 10XUAS-smFP-HA-drep2-sv40. This labeling strategy reports overexpressed, not endogenous, levels of Drep2 protein that appear as punctate signals or small clusters of punctate signals, which were grossly similar to presynaptic sites structures (*Figure 4K and M-O*). Due to the similarity of the labels we were seeking to analyze, we tested whether the U-Net model trained on presynaptic site data could be used to classify postsynaptic receptors as labeled by 10XUAS-smFP-HA-drep2-sv40. Visual inspection indicated that receptor labels at presynaptic site–receptor pairs were commonly smaller than presynaptic sites. Therefore, we reduced the receptor size threshold compared to the presynaptic site size threshold and utilized the post-U-Net processing steps with a 200 pixel minimum size filter on these postsynaptic sites. Upon visual inspection, the classifier worked well. Clusters of Drep2 were grouped together or separated similarly to BRP as anticipated. Therefore, we used the same classifier to identify presynaptic sites and postsynaptic sites labeled by Drep2.

Finally, we used the same strategy to classify putative gap junctions or hemichannels as labeled by innexin 6 (*Figure 5*). Here, the INX6 label appeared in two forms: as punctate signals or as larger clusters or plaques. We used the same U-Net model trained on presynaptic sites with post-U-Net watershed segmentation and a size filter of 200 pixels. This strategy worked well to, preliminarily at least, associate INX6 with segmented neuron masks and classify potential gap junctions.

In principle, this detector should work well on detecting fluorescent punctate signals and clusters labeling other presynaptic, receptor, or gap junction proteins. However, the detector can be readily trained to classify any specific label if sufficient ground truth data can be generated.

We include the trained model used here for classifying synaptic sites, code and instructions to train the classifier, and code and instructions to calculate performance of the classifier (https://github.com/JaneliaSciComp/exllsm-synapse-detector; *Rokicki and Lillvis, 2021*). These components can be run locally or on a compute cluster and can be run independently or as part of several common use workflows described below (https://github.com/JaneliaSciComp/exllsm-circuit-reconstruction).

## Synapse connectivity analysis workflow

With segmented neuron masks and a trained model to classify synaptic sites in hand, the connectivity analysis pipeline can be used. To analyze large ExLLSM imaging volumes, we partitioned the images into 512 × 512 × 512 pixel subvolumes, analyzed these subvolumes in parallel, and then restitched the analyzed results.

We developed four common use analysis workflows that support flexible parameter inputs (*Figure 2—figure supplement 3*). The first workflow quantifies the presynaptic (or postsynaptic) sites in neuron 1 (*Figure 2—figure supplement 3A*). The second workflow quantifies presynaptic sites in neuron 1 and connections from neuron 1 to neuron 2 based on neuron 2 postsynaptic site contact with neuron 1 presynaptic sites. This workflow can be used if presynaptic sites are labeled ubiquitously and postsynaptic sites are labeled specifically in neuron 2 (as with 10XUAS-smFP-HA-drep2-sv40 labeling, e.g., *Figure 4B-F*, *Figure 2—figure supplement 3B*) or if postsynaptic sites are labeled ubiquitously (e.g., with an antibody against a receptor protein) and presynaptic sites are labeled specifically in a neuron (as with STaR-BRP). The third workflow quantifies presynaptic sites in neuron 1 and connections from neuron 1 to neuron 2 based on neuron 1 presynaptic site contact with the neuron 2 membrane (e.g., *Figure 4I-O*, *Figure 6E and F*, *Figure 2—figure supplement 3C*). The fourth workflow quantifies all pre- or postsynaptic sites in a given volume (*Figure 2—figure supplement 3D*).

These workflows are composed of 2–4 stages. At stage 1 of all workflows, pre- and/or postsynaptic label data is classified by the trained U-Net model (*Figure 2—figure supplement 3A–D*). At stage 2 of the fourth workflow (*Figure 2—figure supplement 3D*), classified synaptic sites are run through watershed segmentation and size filtered. At stage 2 of all other workflows, the neuron mask and synaptic site subvolumes are compared; any subvolumes without a neuron mask present are ignored. Subvolumes where the neuron mask is present are analyzed further. Synaptic sites are run through watershed segmentation, size filtered, and assigned to a neuron mask via colocalization (*Figure 2—figure supplement 3A–C*). This colocalization value is flexible – the centroid of the synapse or any percentage of the synapse overlap with the neuron mask can be used. We found that synaptic sites were correctly assigned to neurons if 50% or more of the synaptic site volume overlapped with the neuron mask. Therefore, we used a value of 50% for all data here. At stage 3 of the second and third workflows (*Figure 2—figure supplement 3B and C*), connections were quantified via either neuron 2 postsynaptic site contact with neuron 1 presynaptic sites (*Figure 2—figure supplement 3B*) or neuron 1 presynaptic site contact with neuron 2 (*Figure 2—figure supplement 3C*). Here, we used a colocalization of 0.1% – essentially any contact between pre- and postsynaptic sites was considered a connection. Finally, at stage 4 of the second workflow, the neuron 1 presynaptic sites that contact (0.1% colocalization) neuron 2 postsynaptic sites are identified and quantified (*Figure 2—figure supplement 3B*). The synaptic sites and connections identified each stage are collated into a stage-specific csv file that includes the shape and size of the synaptic site and the position of the site or connection. Additionally, images of the results from each stage are exported in N5 format with options to automatically generate TIFF series and VVD Viewer pyramid files. Examples of how to flexibly utilize each of these workflows or to run steps of each workflow independently are detailed at GitHub (https://github.com/JaneliaSciComp/exllsm-circuit-reconstruction).

## Behavior experiments

For experiments comparing SAG>pC1 connectivity in virgin and mated females (*Figure 4H*), virgin female flies were collected shortly after eclosion. After 2–3 days, individual females were either transferred to a new vial without male flies or to a new vial with male flies. After two additional days, virgin and mated females were dissected, processed for immunohistochemistry, expanded, imaged, and

analyzed to quantify structural connectivity. Virgin and mated females were confirmed as such by the lack or presence of moving larvae in the final vial, respectively.

For pC2l>pIP10 structure–function experiments (*Figure 6*, *Figure 6—figure supplement 1*), naïve males were collected shortly after eclosion and single-housed for 4–6 days before beginning behavior experiments. Crosses and male aging were conducted in the dark on standard media containing 0.4 mM trans-retinal.

Song behavior experiments were conducted in a song recording apparatus described previously (*Arthur et al., 2013*; *Ding et al., 2019*). pC2l neurons were optogenetically activated in isolated males. For CsChrimson activation, constant 635 nm light was applied. Pulse width modulation with a 100 kHz frequency was used to adjust light intensity. A stimulation cycle consisted of 25 s OFF and 5 s ON at the following light intensities: 2.5, 5.3, 8.0, 10.8, and 15.6 $\mu W/mm^2$. This cycle was repeated three times and response were averaged across trials. Pulse and sine song events were annotated manually. Because the only light intensity to reliably elicit song behavior was the maximum 15.6 $\mu W/mm^2$, behavior from this light intensity was used for comparisons in *Figure 6* and *Figure 6—figure supplement 1*.

After optogenetic activation experiments, a 1–2-day-old virgin female was paired with each male and audio was recorded for 10 min in dim blue light. The song generated in these assays was analyzed using SongExplorer (*Arthur et al., 2021*). A convolutional neural network trained for 3,024,000 steps on over 5000 pulse, 3000 sine, 2000 inter-pulse interval, and 2000 other manually annotated events was used to classify song. This trained model exhibited ~80% precision and recall on novel song data from flies not included in the training data set.

Each fly was catalogued and the functional synaptic connection from pC2l and pIP10 was tested the following day via calcium imaging.

## Calcium imaging

Individual flies tested the previous day in behavior experiments were anesthetized by cooling. The brain and ventral nerve cord were removed from the animal and placed into external saline composed of (in mM) 103 NaCl, 3 KCl, 5 N-tris(hydroxymethyl) methyl-2-aminoethane-sulfonic acid, 10 trehalose dihydrate, 10 glucose, 26 $NaHCO_3$, 1 $NaH_2PO_4$, 4 $MgCl_2$, 3 KCl, 2 sucrose, and 1.5 $CaCl_2$ (280–290 mOsm, pH 7.3; components from Sigma-Aldrich). The brain with attached ventral nerve cord was transferred to a chamber (Series 20 Chamber, Warner Instruments) super-fused with external saline (carboxygenated with 95% $O_2$/5% $CO_2$) and held into place via a custom holder.

The GCaMP6f signal was visualized in pIP10 via two-photon imaging with a laser (Chameleon, Coherent) tuned to 920 nm, a Zeiss Examiner Z1 with W Plan-Apochromat 20×/1.0 DIC M27 75 mm water immersion objective, and Zeiss Zen Software. The GCaMP signal was monitored in a single plane for 80 cycles at 391 ms/cycle for a total imaging bout scan time of ~30 s. This 30 s imaging bout consisted of a 5 s baseline, activation of the CsChrimson-tdTomato-expressing pC2l neurons via a constant-on, 5 s 635 nm LED pulse, and 20 s of post-pC2l-stimulation imaging. After a 30 s break, another imaging bout was run. The following pC2l-stimulating 635 nm light intensities were used from low to high: 0.2, 0.5, 0.9, 1.3, 1.9, 2.3, 3.7, 5.3, and 16.6 $\mu W/mm^2$ (pE-4000, CoolLED) through the objective. This stimulation cycle was repeated three times and responses were averaged across trials. Because 8–10% of 635 nm light passes through the *Drosophila* cuticle (*Inagaki et al., 2013*; *Ding et al., 2019*), the 0.2–1.9 $\mu W/mm^2$ irradiances approximately corresponded with the light stimuli given during behavior experiments. Data from the 1.9 $\mu W/mm^2$ stimulus was used in *Figure 6* and *Figure 6—figure supplement 1* because this light intensity corresponded to the light intensity used for optogenetic behavior analyses. A custom band-pass filter (Chroma) allowed constant visualization of the GCaMP6f signal while stimulating with 635 nm light.

ROIs were manually drawn around pIP10 arbors. An additional background ROI was drawn to subtract stimulus noise from the raw GCaMP6f signal. After background subtraction, the baseline data points were averaged to determine baseline fluorescence F0, and the calcium signal was normalized to the baseline and multiplied by 100 ($\Delta F/F0$ %: Fx-F0/F0 * 100). The average response during the 5 s pC2l stimulation was used as the physiological connection value plotted in *Figure 6G* and *Figure 6—figure supplement 1D*. Tissues were immediately fixed after GCaMP imaging and processed through immunohistochemistry, expanded, imaged, and analyzed to quantify anatomical connectivity.

## Irradiance calculations for behavior and calcium imaging experiments

Irradiance was measured using a ThorLabs PM100D Compact Power and Energy Meter with a Console S130C Slim Photodiode Power Sensor. For behavior experiments, the sensor (diameter, 9.5 mm) was positioned in the same location as the arena (diameter, 10.5 mm) directly over the recording chamber microphone. Irradiance was calculated as the raw light power measured divided by the area of the sensor (70.88 mm$^2$).

For calcium imaging experiments, the 635 nm LED stimulus (pE4000, CoolLED) was delivered (with stacked 2.0 and 1.0 neutral density filters in the beam path) through a Zeiss Examiner Z1 with W Plan-Apochromat 20×/1.0 DIC M27 75 mm water immersion objective. The LED beam size was calculated using a beam profiler (WinCamD-UCD12, DataRay) with the sensor placed at approximately the same distance from the objective as the sample during experiments (2 mm). This yielded a 1/e$^2$ beam area of 0.95 mm$^2$. Light power was also measured with the sensor placed 2 mm away from the center of the objective. In an effort to measure the light power of the focused beam and reduce the amount of unfocused or reflected light from being measured by the 70.88 mm$^2$ sensor, a painted black foil sheath was placed over the sensor with an opening for the objective to deliver light. Irradiance was calculated as the raw light power measured divided by the 0.95 mm$^2$ focused beam area.

## Figure preparation

Images and videos were generated in VVD Viewer. Gamma, alpha, and saturation values were adjusted in VVD Viewer to facilitate visualization of overlapping structures. Images were then imported to and assembled in Adobe Illustrator. Scale bars of expanded samples are not adjusted to show pre-expansion size, and thus, show the size of structures after 8× expansion. Microscope objective and fly illustrations in *Figures 2 and 6*, and neural processes in *Figure 2—figure supplement 1* were created with BioRender.com.

## Materials availability statement

All software and code used for data analysis is available at GitHub (https://github.com/JaneliaSci-Comp/exllsm-circuit-reconstruction). Ground truth data used to train the synapse classifier is available at Dryad (https://doi.org/10.5061/dryad.5hqbzkh8b). All genetic reagents are available upon request. The data used to generate the figures and videos in this article exceeds 100 TB. Therefore, it is not practical to upload the data to a public repository. However, all data used in this article will be made freely available to those who request and provide a mechanism for feasible data transfers (physical hard drives, cloud storage, etc.). Documentation for construction of a lattice light sheet microscope can be obtained by execution of a research license agreement with HHMI.

# Acknowledgements

We thank Ramya Kappagantula and Claire Managan for help with generating presynaptic site ground truth data; Damien Alcor and Michael DeSantis for help with imaging; Eric Betzig for input on the LLSM build; Igor Negrashov for mechanical design of the LLSM; Yoshi Aso, Stephan Sigrist, Aljoscha Nern, Yichun Shuai, Glenn Turner, and Gerry Rubin for transgenic fly reagents; the Janelia FlyLight project team for generating the images of unexpanded fly brains; David Ackerman for helpful data analysis discussions and help with image stitching at Janelia; Monique Copeland for help with ExM sample preparation; Gudrun Ihrke for helpful immunohistochemistry discussions; Meng-Hsuan Chiang and Chia-Lin Wu for the INX6 antibody; Ken Carlile, Rob Lines, Habib Bukhari, and Stephan Preibisch for help with data analysis on the Janelia LSF cluster; Sean Murphy for data analysis discussions; Lou Scheffer for L2 data from the 7-column optic lobe FIB-SEM dataset; Stuart Berg and the FlyEM project team for SAG, pC1, and DA1-IPN data from the hemibrain FIB-SEM dataset; Philipp Schlegel and Greg Jefferis for DA1-IPN data from the TEM dataset; Fei Wang for helpful discussions; Henry Ngo for workstation help; David Stern for helpful comments on the manuscript; and Ben Arthur, Emily Behrman, and Elizabeth Kim for help with courtship song recording and analysis.

## Additional information

### Competing interests

Ruixuan Gao, Paul W Tillberg: is a co-inventor on multiple patents related to expansion microscopy. Edward S Boyden: is a co-inventor on multiple patents related to expansion microscopy and is also a co-founder of a company that aims to pursue commercial deployment of expansion microscopy-related technology. The other authors declare that no competing interests exist.

### Funding

| Funder | Grant reference number | Author |
| --- | --- | --- |
| Howard Hughes Medical Institute | | Barry J Dickson |

The funders had no role in study design, data collection and interpretation, or the decision to submit the work for publication.

### Author contributions

Joshua L Lillvis, Conceptualization, Resources, Data curation, Software, Formal analysis, Supervision, Investigation, Visualization, Methodology, Writing – original draft, Project administration, Writing – review and editing; Hideo Otsuna, Xiaoyu Ding, Igor Pisarev, Takashi Kawase, Cristian Goina, John Bogovic, Linus Meienberg, Software, Methodology; Jennifer Colonell, Resources, Methodology, Writing – review and editing; Konrad Rokicki, Ruixuan Gao, Daniel E Milkie, Software, Methodology, Writing – review and editing; Amy Hu, Edward S Boyden, Methodology; Kaiyu Wang, Resources; Brett D Mensh, Writing – review and editing; Stephan Saalfeld, Software, Supervision, Methodology; Paul W Tillberg, Supervision, Methodology, Writing – review and editing; Barry J Dickson, Conceptualization, Supervision, Funding acquisition, Writing – original draft, Project administration, Writing – review and editing

### Author ORCIDs

Joshua L Lillvis http://orcid.org/0000-0002-6235-8759
Hideo Otsuna http://orcid.org/0000-0002-2107-8881
Konrad Rokicki http://orcid.org/0000-0002-2799-9833
Cristian Goina http://orcid.org/0000-0003-2835-7602
John Bogovic http://orcid.org/0000-0002-4829-9457
Linus Meienberg http://orcid.org/0000-0001-6793-1422
Edward S Boyden http://orcid.org/0000-0002-0419-3351
Stephan Saalfeld http://orcid.org/0000-0002-4106-1761
Paul W Tillberg http://orcid.org/0000-0002-2568-2365

### Decision letter and Author response

Decision letter https://doi.org/10.7554/eLife.81248.sa1
Author response https://doi.org/10.7554/eLife.81248.sa2

## Additional files

### Supplementary files

• Supplementary file 1. Genotype, neurons, reporters, age, sex, antibodies, antibody concentrations, and transgenic fly source for all data and figure panels.

• MDAR checklist

### Data availability

All software and code used for data analysis is available at Github (https://github.com/JaneliaSci-Comp/exllsm-circuit-reconstruction, copy archived at swh:1:rev:2b6ffd97ef80d31d75cdc1acb-f227aedab1cb409). Ground truth data used to train the synapse classifier is available at Dryad (https://doi.org/10.5061/dryad.5hqbzkh8b). All genetic reagents are available upon request. The data used to generate the figures and videos in this manuscript exceeds 100TB. Therefore, it is not practical to upload the data to a public repository. However, all data used in this paper will be made freely

available to those who request and provide a mechanism for feasible data transfers (physical hard drives, cloud storage, etc.). Documentation for construction of a lattice light-sheet microscope can be obtained by execution of a research license agreement with HHMI.

The following dataset was generated:

| Author(s) | Year | Dataset title | Dataset URL | Database and Identifier |
|---|---|---|---|---|
| Lillvis JL | 2022 | Ground truth data used to train the synapse classifier used in Lillvis et al., 2022 for ExLLSM circuit reconstruction | https://doi.org/10.5061/dryad.5hqbzkh8b | Dryad Digital Repository, 10.5061/dryad.5hqbzkh8b |

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
