## [Editor Report]

This article introduces a new light microscopy pipeline for imaging and fast reconstruction of the synaptic connections of individual neuronal types in the fruit fly and for correlated investigation of circuit structure, function, and behavior in the same animal. Because of its speed and accessibility, this approach enables the mapping of selected neuronal circuits of multiple animals across different conditions and behavioral states, thus filling an important gap in brain research.

---

## [Decision Letter]

**Decision letter after peer review:**

Thank you for submitting your article "Rapid reconstruction of neural circuits using tissue expansion and light sheet microscopy" for consideration by *eLife*. Your article has been reviewed by 3 peer reviewers, and the evaluation has been overseen by a Reviewing Editor and Ronald Calabrese as the Senior Editor. The reviewers have opted to remain anonymous.

Essential revisions:

The authors should discuss in more detail the limitations of their technology and its transferability to other species with larger brains or species where no advanced genetic tools are available yet. In particular, it would be helpful to give the reader a clear picture of the use cases where it's not expected to work. Reviewer #3 listed a series of points that you might want to clarify in the discussion of your results. No additional data is required to address the criticisms raised by the Reviewers.

*Reviewer #2 (Recommendations for the authors):*

The authors could either extend the Discussion or may want to add some more columns to Figure 1. For example, they could contrast dense and sparse reconstruction between LM/EM e.g. 'allows dense reconstruction', which is possible by EM but not (yet) by ExLLSM. Another fair comparison to make is if a method 'requires neuron-specific genetic tools', which is also a limitation of ExLLSM but not EM.

*Reviewer #3 (Recommendations for the authors):*

More detailed questions/suggestions:

1) In Figure 2, label what the different colors mean in overview (like, green: neurons of type 1, magenta: neurons of type 2). Do not reuse the same colors in panes M and N. In pane Q, the axes are very hard to read, zoom into region between 0.8 and 1.

2) Line 583, what do you mean by "the Huang or Li method"?

3) The Methods section keeps mentioning "TIFF series" as if it's important to distinguish those from the VVD format or N5 (lines 579-586 and many others). I do not quite understand the workflow then, do you always keep the data in a TIFF series as well? I thought you had 10s of Terabytes, why duplicate?

4) Line 651, why do you need to use the Fiji N5 viewer and then switch to VVD again for semi-automatic segmentation? Does VVD not support cropping?

5) Video 1 has an embedded "legend" saying that red is presynaptic sites. What does this refer to, the video is showing multi-colored segmented synapses?

---

## [Author Response]

Essential revisions:The authors should discuss in more detail the limitations of their technology and its transferability to other species with larger brains or species where no advanced genetic tools are available yet. In particular, it would be helpful to give the reader a clear picture of the use cases where it's not expected to work. Reviewer #3 listed a series of points that you might want to clarify in the discussion of your results. No additional data is required to address the criticisms raised by the Reviewers.

We have added text to the discussion to illustrate what is needed for the application of our tools as they currently stand and include examples of organisms that may be well-suited to our approach (lines 363-367). Already included in the discussion are points about the current limitations and how future development may overcome these limitations. In particular, lines 374-379 explicitly discuss the need to improve labeling continuity and develop automated single neuron segmentation methods in order to analyze single neurons or densely labeled samples as opposed to analyzing neuron types and sparsely labeled samples as we have done here. We have made additional additions to the text elsewhere in service of these concerns and have documented these changes below.

Lines 363-372: Accordingly, these tools should be well-suited for any organism in which neurons can be reliably labeled and tissue can be sufficiently expanded. This includes other *Drosophila* species, nematodes, rodents, zebrafish, and organoids, for each of which genetic tools to label specific neurons are available and expansion microscopy protocols have been established (Freifeld et al., 2017; Gao et al., 2019; Yu et al., 2020; Rodriguez-Gatica et al., 2022). The development of such genetic tools and expansion protocols in other species will extend the applicability of this approach across taxa. Furthermore, the convolutional neural network models for synapse and neuron segmentation are classifiers of high signal punctate and continuous structures, respectively. As such, the models may already work well for segmenting similar structures from other species or microscopes. If not, these models can be retrained with a suitable ground truth data set and the entire computational pipeline can be applied to these new systems.

Reviewer #2 (Recommendations for the authors):The authors could either extend the Discussion or may want to add some more columns to Figure 1. For example, they could contrast dense and sparse reconstruction between LM/EM e.g. 'allows dense reconstruction', which is possible by EM but not (yet) by ExLLSM. Another fair comparison to make is if a method 'requires neuron-specific genetic tools', which is also a limitation of ExLLSM but not EM.

Please see our response to the Essential Revisions (for the authors) section above.

Reviewer #3 (Recommendations for the authors):More detailed questions/suggestions:(1) In Figure 2, label what the different colors mean in overview (like, green: neurons of type 1, magenta: neurons of type 2). Do not reuse the same colors in panes M and N. In pane Q, the axes are very hard to read, zoom into region between 0.8 and 1.

We have made the suggested changes.

(2) Line 583, what do you mean by "the Huang or Li method"?

References for each thresholding method were added.

(3) The Methods section keeps mentioning "TIFF series" as if it's important to distinguish those from the VVD format or N5 (lines 579-586 and many others). I do not quite understand the workflow then, do you always keep the data in a TIFF series as well? I thought you had 10s of Terabytes, why duplicate?

There is no need to duplicate data in different formats at any stage of any workflow and N5 can be used for almost every step. However, the output of VVD Viewer semiautomatic segmentation is a TIFF series (with no other export options). We use these TIFF series for the Fiji-based steps of the post-VVD neuron segmentation processing workflow (ROI cropping, thresholding, 3D component connecting). We therefore reference the TIFF series when discussing these steps. We additionally discuss the ability to convert to or from TIFF series as this may be desired or required for analyses outside of our workflows.

(4) Line 651, why do you need to use the Fiji N5 viewer and then switch to VVD again for semi-automatic segmentation? Does VVD not support cropping?

VVD does not currently support cropping in z or cropping of VVD or N5 file formats X, Y, or Z. However, N5 format is the output of the stitching process, and it is straightforward to take small crops using the N5 viewer. We also generated a Fiji macro to crop TIFF series which was used to reduce the size of the volume used for analysis after neuron segmentation (see ROI cropping starting on line 647).

(5) Video 1 has an embedded "legend" saying that red is presynaptic sites. What does this refer to, the video is showing multi-colored segmented synapses?

This refers to the multi-color segmented synapses. Red was used arbitrarily -- there is no option to make multi-color labels in VVD Viewer (where the video was generated). We have changed the label color here and indicate that segmented synapses are multicolor in the legend.